# A Comparative Study on the Direct and Pulsed Current Electrodeposition of Cobalt-Substituted Hydroxyapatite for Magnetic Resonance Imaging Application

**DOI:** 10.3390/ma12010116

**Published:** 2018-12-31

**Authors:** Wei-Chun Lin, Chun-Chao Chuang, Pin-Ting Wang, Cheng-Ming Tang

**Affiliations:** 1Institute of Oral Science, Chung Shan Medical University, Taichung 40201, Taiwan; tukust94114wenny@gmail.com (W.-C.L.); she07866@gmail.com (P.-T.W.); 2Department of Medical Imaging and Radiological Sciences, Chung Shan Medical University, Taichung 40201, Taiwan; jimchao@csmu.edu.tw; 3Chung Shan Medical University Hospital, Taichung 40201, Taiwan

**Keywords:** cobalt-substituted hydroxyapatite, pulsed electrode position, magnetic resonance imaging, antibacterial, anti-inflammatory

## Abstract

Hydroxyapatite has excellent biocompatibility and osteo-conductivity and, as the main inorganic component of human bones and teeth, is commonly used for bone repair. Its original characteristics can be changed by metal ion substitution. Cobalt ions can act as hypoxia-inducible factors and accelerate bone repair. At the same time, cobalt has paramagnetic properties and is often used in the study of medical imaging and target drugs. Through the introduction of cobalt ions, the unique hydroxyapatite has better biological activity and positioning of medical images. Herein, cobalt-substituted hydroxyapatite (CoHA) was synthesized on the surface of a titanium plate by electrochemical deposition and changes in the power output mode to explore the impact on CoHA. Electrochemical deposition with a pulse current significantly improved the productivity and uniformity of CoHA on the surface of titanium. CoHA show paramagnetic characteristics by a superconducting quantum interference device (SQUID). Resulting smaller particle size and circular morphology improves the magnetic strength of CoHA. Magnetic resonance imaging (MRI) of CoHA showed significant image contrast effect at low concentrations. The calculated particle relaxation rate was higher than other common MRI contrast agents. Biocompatibility of CoHA powder was evaluated using the human osteosarcoma cell line (MG63) which confirmed that CoHA is not cytotoxic and can promote cell growth and extracellular matrix mineralization. With the release of cobalt ions, CoHA was found to be significantly good in repression *E. coli* indicating about than 95% reduction in bacterial growth. The as-synthesized CoHA has a low degree of crystallinity, highly sensitive image contrast effect, and good bioactivity, and may have potential applications in bone repair and MRI.

## 1. Introduction

Patients with defects due to trauma or osteoporosis must rely on fillers to make repairs. Currently, autologous bone graft, allogeneic bone graft and xenograft are widely used in bone filler materials. However, an autologous bone graft is limited by the disease incidence of donor and donor sites [1] and may incur high costs due to development of complications [2]. Allogeneic bone grafts and xenografts might lead to immune rejection and infectious diseases [3]. Therefore, many recent studies have focused on the development of calcium phosphate bone graft material [4,5]. Hydroxyapatite (HA) is the common form of calcium phosphate bone graft [6]. HA is the main inorganic component of bones and teeth of vertebrates. Bases on the stability and flexibility of the HA crystal structure, calcium ions on HA can be replaced with divalent metal cation such as iron [7], magnesium [8], silver [9], cobalt [10] etc. Due to cobalt ion being a hypoxia-mimicking agent, it can activate the hypoxia inducible factor-1 (HIF-1α) in mesenchymal stem cells and subsequently activate HIF-1α target genes including vascular endothelial growth factor (VEGF) erythropoietin (EPO) [11,12,13,14]. Previous literature showed that cobalt substituted hydroxyapatite (CoHA) designed on the nano-scale can enhance osteogenesis in vivo [15]. Cobalt-substituted hydroxyapatite has paramagnetic properties [16] and can be developed as a contrast agent in magnetic resonance imaging (MRI) [17]. The superior contrast agents must have low toxicity, excellent chemical stability, high magnetic moment and ability to bind ligands to particles [18]. The application of CoHA should reduce the negative effects of currently used contrast agents on the human body. Traditionally, hydroxyapatite can be synthesized by many methods such as hydrothermal [19,20,21], wet precipitation [7,8,10,22] and electrochemical deposition [23,24,25,26,27,28,29,30,31]. In particular, electrochemical deposition can be achieved at lower working temperature and using simpler equipment than other methods. On the other hand, the HA coating by electrochemical deposition formed is more uniform [32] and has higher bond strength [15]. In our previous study, CoHA was successfully synthesized by electrochemical deposition procedure [33]. Electrochemical deposition can be divided into direct current (DC) power supply and pulsed current (PC) power supply according to the current supply mode. During the reaction, the instantaneous energy associated with a pulse current is higher than that of a direct current, leading to a reduction of metal ions at an extremely high over-potential. When the current is turned off, the discharge ion concentration near the cathode is restored to its initial value and concentration polarization disappears. This phenomenon for the next pulse cycle is conducive to the deposition of the surface. The purpose of this study is to synthesize CoHA via electrochemical deposition. Using the pulse current supply mode, the ion concentration near the cathode recoverd its equilibrium value when the current was turned off, thus facilitating adsorption and generate of HA. The effects of different current supply modes on the morphology, magnetic propertiesand biodegradability were also compared in this study. The bioactivity of the CoHA powder was evaluated by examining the behavior of osteoblasts using MTT (3-(4,5-Dimethylthiazol-2-yl)-2,5-diphenyltetrazolium bromide) and Alizarin Red S assays. An evaluation of the antibacterial effects of CoHA using *E. coli* was carried out. Ultimately, this study aims to optimize CoHA production, to promote the repair of bone defects through release of cobalt ions and to provide marking and tracking in clinical MRI images.

## 2. Materials and Methods

### 2.1. Synthesis of Cobalt-Substituted Hydroxyapatite by Electrochemical Deposition

A commercial titanium sheet (99.9%, Opetech materials, Hsinchu, Taiwan) and 304 stainless-steel plate (Extra pure, Taichung, Taiwan) was cut into 80 mm × 40 mm × 1 mm. The samples were washed with acetone, ethanol and deionized water for 10 min using an ultrasonic shaker and removed at room temperature to dry. The electrolytic solution used 42 mM calcium nitrate (Shimada chemical works, Tokyo, Japan) and 25 mM ammonium dihydrogen phosphate (Showa chemical industry, Tokyo, Japan) in deionized water [34]. The pH of the electrolyte was adjusted at 3.52 using hydrochloric acid and tris(hydroxymethyl)aminomethane (Tris). Finally, 7.9 mM Cobalt chloride (Shimada chemical works, Tokyo, Japan) was added and stirred until it completely dissolved. Titanium piece as a cathode, 304 stainless steel plate as an anode. Were used direct current (DC) and pulse current (PC) for electrochemical deposition (Figure 1). The temperature of the electrolyte solution was controlled at 55 °C and the voltage was 5.5 V for electrochemical deposition. There are three groups in this experiment, for the DC-CoHA (DC-type electrode position), PC_1_-CoHA (pulsed electrode position same time as the DC), and the PC_2_-CoHA (pulsed electrode position with the same power as the DC), respectively (Table 1). After deposition, CoHA was cleaned using deionized water and dried at room temperature. Finally, the powder on the titanium piece was scraped off to acquire the CoHA particles.

### 2.2. Surface Characterization

The microstructures of the surface of the powder were observed by a field-emission scanning electron microscope (FE-SEM) (JSM-7610F, JEOL, Tokyo, Japan). Element composition on the powders surface was measured by energy-dispersive X-ray spectroscopy (EDS, JEOL, Tokyo, Japan). The Fourier transform infrared spectrometer (FTIR) (Vertex80v, Bruker, Billerica, MA, USA) was used to study the surface functional groups of materials. The scanning range of this study is 400–4000 cm^−1^ and the number of scans is 200 times. The phase composition of the CoHA powders was determined by X-ray diffractometer (XRD) (Miniflex II, Rigaku, Tokyo, Japan) using CuKα radiation (λ = 1.54 Å). The scanning conditions for this study were 4 °/min and the scanning range was 10°–70°. The crystal size of hydroxyapatite is calculated by the following formula, Xs = 0.9λ/FWHM X cosθ, Xs is the grain size (nm), λ is the CuKα radiation, FWHM (full width at half maximum) is the half width of the diffraction peak, θ for the diffraction peak angle. The crystallinity of hydroxyapatite is calculated by the following formula: Xc = (K/β)^3^; K is a constant of 0.24 [35].

### 2.3. Internal Ion Composition

Determination of the overall elemental composition in the CoHA was undertaken by inductively coupled plasma optical emission spectrometry (ICP-OES) (Optima 8300, Perkin Elmer, Waltham, MA, USA). The amount of powder was weighed, 1 mL of 65% nitric acid was added so that the sample completely dissolved, and then it was filtered through 0.45 μm filter and the concentration range and dilution of the calibration line was set. 

### 2.4. Magnetic Analysis

Magnetic properties of CoHA was obtained using superconducting quantum interference devices magnetometer (SQUID) (MPMS5, Quantum Design, San Diego, CA, USA) in an applied magnetic field of −10,000 Oe to 10,000 Oe at 37 °C. 

### 2.5. In Vitro Biodegradation

Each sample take 0.1 g was immersed in 10 mL of phosphate buffered saline (PBS), placed in a constant temperature water bath and the temperature was maintained at 37 °C for four weeks and the pH was measured daily. The PBS composition has KCL, KH_2_PO_4_, NaCl and Na_2_HPO_4_. The samples were soaked for 7 days and the release of ions was measured by ICP-OES.

### 2.6. Free Radical Scavenging Effects 

The free radical scavenging activity was evaluated using 2,2-diphenyl-1-picrylhydrazyl (DPPH) free radical [36,37]. One mL of distilled water (as control group) or 1 mL of deionized water containing 0.01 g CoHA, was added to 3 mL of 32-μM DPPH free radical in methanol and left to stand for 90 min at room temperature. Absorbance of the reaction mixture was then measured at 517 nm using an ultraviolet–visible spectrophotometer (UV/VIS) (Helios Zeta, Thermo, Waltham, MA, USA). The free radical scavenging effect is determined by the following equation: scavenging ratio (%) = [1 − (absorbance of test sample/absorbance of control)] × 100%.

### 2.7. In Vitro Magnetic Resonance Imaging (MRI) Examination

MRI tests were performed on a 1.5 T MRI scanner (Signa Horizon LX, GE Healthcare, Chicago, IL, USA). A certain amount of HA and CoHA powders were dispersed in a gelatin (Sigma Aldrich, St. Louis, MO, USA) with different concentrations and then poured into 5 mL distilled water. The sample and gelatin solution were mixed thoroughly while at 70 °C. The gelatin mixture was allowed to cool to room temperature. The T_1_-weighted images were acquired using spin echo imaging sequencing with the following parameters: matrix size = 256 × 256, field of view = 180 mm × 180 mm, slice thickness = 5 mm, echo time = 26 ms, repetition time = 100 ms, number of acquisitions = 2. Color image processing was performed using ImageJ (1.51K, National Institutes of Health, Bethesda, MD, USA). The contrast-enhancing efficacy of CoHA and commercial HA were determined by its relaxation coefficient (*r*_2_). The *r*_2_ is calculated by the following formula (Equation (1)) [38]:(1)$$\frac{1}{T_{2}}=\frac{1}{T_{2}^{0}}+r_{2}C$$
where $T_{2}$ is the observed relaxation time in the presence of CoHA, $T_{2}^{0}$ is the relaxation rate of pure gelatin and C is the concentration of cobalt ion.

### 2.8. Biocompatibility 

We used a mouse-derived established cell line (L929) of fibroblasts and human osteosarcoma cell line (MG63), maintained in Dulbecco’s modified eagle medium (DMEM) supplemented with 10% fetal bovine serum (FBS) (Biological Industries, Cromwell, CT, USA) at 37 °C in a 5% CO_2_ incubator (310, Thermo Fisher Scientifc, Waltham, MA, USA) in this study in order to investigate the effect of ions released by CoHA on cells. We used the CoHA extract for testing. Using 0.1 g of three CoHA powders each immersed in 1 mL of deionized water and soaked for 7 days at 37 °C. Moreover, we used cobalt ion standard solution (AccuStandard, New Haven, CT, USA), diluted to 1 and 10 ppm, as a positive control followed by centrifugation, aspiration of the supernatant, and removal of the liquid using a freeze dryer (FDU-1200, Tokyo Rikakikai, Tokyo, Japan). Then 10 mL of DMEM was added to each sample and filter with a 0.22 μm sterile filtration apparatus as a culture medium for cell test. L929 and MG63 cells were seeded in 24 well at 5 × 10^4^ cells/well, cultured for 18 h and replaced with extract. Finally, it was cultured in a 37 °C, 5% CO_2_ cell incubator for 24, 48 and 72 h. The biocompatibility was evaluated by the colorimetric MTT (3-(4,5-Dimethylthiazol-2-yl)-2,5-diphenyltetrazolium bromide, MTT) assay. The absorbance (O.D.) of the wavelength of 563–650 nm was read with an enzyme-linked immunosorbent assay (ELISA) reader (Sunrise, Tecan, Männedorf, Switzerland). The biocompatibility was expressed as the percentage of compared to that control tissue culture plate (TCP).

### 2.9. Extracellular Matrix Mineralization

Alizarin Red S (ARS) (Sigma Aldrich, St. Louis, MO, USA) is an orange-yellow needle-like crystal, alizarin sulfonate sodium salt. With calcium salt to chelate the formation of orange-red deposition complex, the detection of calcium deposition in cultured cells. Using 2% ARS, pH between 4.1–4.3. MG63 cells (1 × 10^4^ cells/well) were cultured with the extract for 3 and 7 days, then washed 2–3 times with PBS and fixed with 4% paraformaldehyde for 10 minutes. After removal, they were rinsed with deionized water once, add ARS, and reaction for 15 minutes at room temperature. Ultimately, deionized water was used to wash them 2–3 times. The results of staining were observed by optical microscope. In the quantitative analysis, the ARS on the specimen after washed with distilled water was dissolved in 0.2 M NaOH/methanol (1:1) to measure the optical density at 620 nm [12].

### 2.10. Antibacterial

The CoHA were investigated against *E. coli* as model Gram-negative bacteria by the colony plate count method in order to quantify the bacterial effect of our system. The *E. coli* were prepared from fresh brain heart infusion (BHI, Becton Drive, Franklin Lakes, NJ, USA) and incubated at 37 °C for 24 h. The BHI containing *E. coli* was diluted to 10^−3^ of its original concentration, and 1 mL of bacteria liquid was extracted and added a fixed weight (6 mg) of CoHA samples, placed into a centrifuge tube and cultivated for 18 h. After the samples were removed, 100-μL bacterial solution was extracted and applied on the BHI Agar (Becton, Dickinson and Company, Houston, TX, USA) petri dish before being cultured for 24 h at 37 °C. Finally, the colonies were counted, and the results were expressed as percentage reduction rates of bacteria number = [α × 10^5^], where α is the number of bacterial colonies.

### 2.11. Statistical Analysis

All data from the average of the three repeat samples ± standard deviation. Data were calculated using JMP14 software (JMP^®^14.1.0, Cary, NC, USA). One-way analysis of variance (ANOVA) was used to examine the differences between groups using the Tukey HSD multiple comparison. *p* < 0.05 is considered to be significant.

## 3. Results and Discussion

### 3.1. Surface Characterization

The CoHA synthesized using different current supply modes were denoted as DC-CoHA, PC_1_-CoHA and PC_2_-CoHA. After cleaning and drying, the deposit appeared lilac in color due to the presence of cobalt ions (Figure 2). The distribution of apatite deposits was analyzed using imaging software (Image J), indicated by the red area observed in Figure 2A. The PC power supply produced better CoHA deposition on the cathode metal plate than DC, because the ions provided by the intermittent power supply mode when pulse current is applied reach an equilibrium content when the current is turned off. The surface morphology of CoHA was observed using FE-SEM (Figure 2B). All groups of CoHA particles were spherical and densely packed together. In 2014, Gopi et al. reported that the spherical morphology HA is conducive to cell attachment and proliferation [39]. Therefore, the shape of CoHA has a positive effect on cell growth. FE-SEM images were analyzed using Image-Pro Plus software. The calculated particle size descended in the order: PC_2_-CoHA, PC_1_-CoHA and DC-CoHA (Table 2). The surface elemental composition of CoHA was analyzed using energy-dispersive X-ray spectroscopy (EDS). The DC-CoHA surface contained the highest amount of elemental cobalt (3.4 at %), followed by PC_2_-CoHA (2.9 at %) and PC_1_-CoHA (2.2 at %). 

The FTIR spectra of CoHA synthesized using different current modes are shown in Figure 3. The absorption peaks at wave numbers of 564.1 cm^−1^, 603.6 cm^−1^, and 1043 cm^−1^ represent ${\mathrm{PO}}_{4}^{3-}$, while those at 1381.1 cm^−1^ and 1647 cm^−1^ represent C=O [40]. The peaks at 873.0 cm^−1^, 1423.3 cm^−1^ and 1483 cm^−1^ represent type B carbonate bands [41,42,43]. One study has indicated that ${\mathrm{CO}}_{3}^{2-}$ can promote the degradation of HA, provides calcium and phosphorus ions. Which can be adsorbed during bone remodeling, thereby improving bone growth [44]. The peak at 3484 cm^−1^, representing ${\mathrm{OH}}^{-}$, are observed in three powder groups DC-CoHA, PC_1_-CoHA, and PC_2_-CoHA (Figure 3B), among which DC-CoHA showed the largest peak shift. This could be caused by the addition of CoCl_2_, which combined with ${\mathrm{OH}}^{-}$ to observed peak shift. The HA component of the CoHA was confirmed via FTIR analysis [35,45].

The X-ray diffraction analysis results of the powder crystal structure is shown in Figure 4. These results were compared with literature; diffraction peaks of HA crystals were observed at 26.01° (002), 32.04° (211), 39.68° (310), 46.81° (222) and 49.74° (123) [46,47], and the diffraction peak of Co_3_O_4_ was observed at 18.8° (111) [16,48], which confirmed the presence of the CoHA structure. Since none of the CoHA powder are calcined, all the samples belonged to the low crystallinity range. Despite this, the PC_1_-CoHA and PC_2_-CoHA samples showed larger half widths and lower peaks. The CoHA produced by the display of the pulse current supply mode has more structures that are amorphous. However, materials with high crystallinity cannot degrade easily, and in bone tissue engineering, accelerated degradation of HA is preferred during bone repair to facilitate the absorption of osteoblasts and rapid recovery of bone defects [49].

### 3.2. Internal Ion Composition

The results calculated based on inductively coupled plasma optical emission spectrometry (ICP-OES) analysis are shown in Table 3. After cobalt ion substitution, the Ca + Co/P ratio was calculated to be 1.67–1.72, similar to the Ca/P ratio of human bone (1.67). The cobalt ion contents for the three CoHA powder groups were 13–14%, slightly higher than the initial electrolyte concentration (10%). While DC-CoHA had the highest surface cobalt ion content according to the EDS surface elemental analysis, the ICP-OES analysis of three groups showed similar internal cobalt substitutions (14%). This indicates that CoHA synthesized using DC power supply has higher surface cobalt ion substitution, while PC power supply results in higher internal cobalt ion substitution.

### 3.3. Magnetic Analysis

The hysteresis curve of CoHA was measured using a superconducting quantum interference magnetometer (Figure 5). According to literature, a negative slope indicates a diamagnetic hysteresis curve for pure HA [16]. In this study, a positive slope was observed for CoHA, corresponding to a paramagnetic hysteresis curve. Coercivity (*Hc*), saturation magnetization (*Ms*), and remanent magnetization (*Mr*) were calculated from Figure 5 (Table 2). The results show that DC-CoHA has the highest magnetism, followed by PC_1_-CoHA and PC_2_-CoHA. The magnetic properties of a material can be affected by many factors, such as size, structure, surface morphology, crystallinity, and defects. The DC-CoHA surface contains relatively more cobalt ions and, therefore, has higher magnetism. On the other hand, the magnetic properties of PC_1_-CoHA and PC_2_-CoHA can be explained by their particle morphologies and sizes (Table 2). PC_1_-CoHA particle size is smaller thanPC_2_-CoHA. Therefore, these two powder groups showed different levels of magnetism. The results are similar to that reported in literature, i.e., smaller particles have higher magnetism [38,50]. In 2014, Kramer et al. [10] synthesized CoHA powder using ion exchange and wet methods. Their magnetic analysis results showed that the *Ms* of CoHA synthesized by ion exchange and wet methods were 0.2 emu/g and 0.1 emu/g, respectively. In 2015, Sarath Chandra et al. [16] reported that CoHA synthesized by combining hydrothermal and microwave irradiation techniques had an *Ms* value of up to 0.06 emu/g. Compared to other synthesis methods, the CoHA synthesized via electrochemical deposition had significantly higher magnetism, with the highest *Ms* value detected herein being 0.9 emu/g. The SQUID results confirmed the substitution of a small number of cobalt ions in the HA lattice, resulting in the generation of a paramagnetic material. In addition, the low crystallinity of the three kinds of CoHAs in this study belongs to amorphous materials. Most amorphous materials show that hysteresis is unusual. However, Erica Kramer et al. used ion exchange and wet synthesis to synthesize CoHA. The authors point out that the two CoHAs have very low crystallinity due to the addition of Co, and it shows that they belong to the amorphous material [10]. However, the results of the magnetic analysis showed that the magnetization of CoHA showed a positive slope indicating a paramagnetic property. Therefore, although the crystallinity of the material is very low in this study, CoHA is superparamagnetic due to the presence of cobalt ions.

### 3.4. In Vitro Biodegradation

To simulate the environment in the human body, samples were soaked in PBS at 37 °C for four weeks. The results of a short-term dissolution test in Figure 6A showed that when three kinds of CoHA were immersed to the PBS, respectively, the anion and cation will be released in the initial period. However, anion (phosphorus) is released faster than cation (calcium, cobalt) lead to increase of pH value. Then, cation release slows down the increase of pH value. This tendency is consistent with the literature [10], slow dissolution of hydroxyapatite and increase of hydroxy group in solution leads to an increase of pH value. Consequently, when the pH value changes, it means that there are more calcium and cobalt ions released. Therefore, CoHA prepared by the PC method can obtain a good degradation effect compared with the DC method. In particular, PC1-CoHA has a significant difference at 24 h. In addition, the pH rises a lot before 24 h, followed by a slow rise and remaining at 8.97 after day 15. The originally neutral PBS solutions for all the groups turned alkaline after four weeks of soaking due to the dissociation that occurred after soaking. The results of CoHA powder after soaking in PBS for seven days are shown in Table 3. The release concentration of cobalt ions was 1.45–1.86 ppm. The release of cobalt ions from PC_1_-CoHA and PC_2_-CoHA is higher than DC-CoHA, thus affecting the pH bias towards neutrality. Due to low crystallinity, the PC_1_-CoHA powder degraded faster. Therefore, CoHA prepared by the PC method can obtain a good degradation effect compared with the DC method. In particular, PC_1_-CoHA has a significant difference at 24 h. At the same time, the literature points out that increasing the concentration of calcium and phosphate ions in the bone defect can effectively promote the growth of bone [51,52]. 

### 3.5. Free Radical Scavenging Effects 

Figure 7 shows the rates of free radical scavenging. HA and cobalt standard solution were used as a control group. The results show that the free radical scavenging ratio of the three groups of CoHA samples mainly comes from cobalt ions. In particular, DC-CoHA had the highest free radical scavenging ratio of 50% among the samples, because the surface of DC-CoHA powder had more cobalt ions, allowing more combinations with free radicals. Free radicals are produced during biological differentiation and defense; however, too many free radicals may cause excessive oxidation. Antioxidants can be used to maintain the normal growth of organisms by capturing harmful free radicals; thus, they play an important role in this balance. A previous study showed that the anti-oxidative properties of Ag nanoparticles in waterborne polyurethanes can reduce both in vitro and in vivo inflammations [53]. Therefore, the free radical scavenging capability of CoHA can induce anti-inflammatory effects in cells, reducing inflammatory reactions and promoting bone repair at defect sites.

### 3.6. In Vitro MRI Examination

Due to the structural differences of human tissues, the differences between tumor and normal tissue can be detected using MRI, and the location of therapeutic agents can also be tracked simultaneously. MRI has many advantages: it has extremely high imaging flexibility, does not have apparent detrimental effects on patients, enables the generation of high-resolution images with excellent contrast for different tissues, provides physiological parameters, and facilitates the acquisition of unique clinical information [54,55]. However, metal implants or fillers used for bone defects often interfere with clinical MRI imaging judgment due to the inherent characteristics of the materials [56]. A previous study, which used cobalt ferrite/graphene oxide nanocomposites (CoFe_2_O_4_/GO) as a carrier, showed a high MRI relaxation rate, indicating its great potential for application in clinical MRI [17]. In this study, commercial HA was used as the control group for the MRI test. To verify the diagnostic potential of CoHA, T_2_-weighted MRI images were obtained at different sample concentrations at room temperature (Figure 8). Figure 8A shows that the signal intensity of the MRI image increases with increasing concentration of cobalt ions, and a clear boundary is also shown in the image. However, no significant changes are observed in the image for pure HA. Figure 8B shows the results of the relaxation rate, *R*_2_ (1/*T*_2_). A linear increase was observed for all samples, indicating the generation of a *T*_2_-weighted spin-echo sequence in the MRI of CoHA. The higher values of *R*_2_ achieved a greater contrast effect. The relaxation rates (*r*_2_) is obtained from the curve of *R*_2_ versus, and is a standardized contrast enhancement index [57]. The CoHA *r*_2_ measured for different cobalt ion concentrations are shown in Figure 8C. DC-CoHA has the highest relaxation rate 283.4 mM^−1^s^−1^, followed by PC_1_-CoHA (227.8 mM^−1^s^−1^) and PC_2_-CoHA (223.3 mM^−1^s^−1^). Although CoHA synthesized using pulse current has a lower relaxation rate than that using direct current, the *r*_2_ was still significantly higher than the contrast agent ferrite ethanol (91 mM^−1^s^−1^) [38] and other cobalt-containing nanoparticles [17,58,59]. MRI contrast agents with high *r*_2_ can drastically shorten the T_2_ relaxation time at relatively low concentrations [38,58]. Since the surface of DC-CoHA may be attached with more cobalt ions (with unpaired electrons, it is easy to disturb the magnetic field), this produces higher magnetic properties (Ms), resulting in higher relaxation rates (*r*_2_) than PC_1_-CoHA and PC_2_-CoHA (Figure 8C). At the same time, together with the literature, the results show that the relaxation rate *r*_2_ is proportional to the saturation magnetization [38,60]. Therefore, DC-CoHA has a higher relaxivity *r*_2_. The MRI test confirmed that CoHA had relatively high relaxation rates at low concentrations, allowing it to be used for localization in T_2_ imaging applications like MRI markers (gold seeds) [61,62,63]. In future, CoHA may be used as a filler for bone defects, and clinicians may be able to determine the progress of bone tissue repair and track the location of CoHA using MRI.

### 3.7. Biocompatibility

The L929 and MG63 cell lines were cultured in the CoHA extract solutions. The MTT assay was used to evaluate the biocompatibility at different time intervals (Figure 9). A tissue culture plate (TCP) was used as the control group. After culturing the L929 cell line for 72 h, no significant differences were observed when comparing the three CoHA groups with 1 ppm cobalt ions and the TCP. However, a sharp drop in cell viability was observed when the cobalt ion concentration was increased to 10 ppm. During the initial period of MG63 culture (24 h), a significant improvement in cell viability was observed for PC_2_-CoHA and the 1-ppm cobalt ion groups. After 72 h of culture, cell viability for PC_1_-CoHA was significantly higher than the other groups, and was 16% higher than the TCP control group. PC_2_-CoHA also had significantly higher cell viability than the control group. The two cell lines show that cobalt ions did not significantly promote proliferation of fibroblasts, whereas the proliferation of human osteosarcoma cells was significantly improved at a cobalt ion concentration of 1–2 ppm. Cell viability decreased when the cobalt ion concentration was increased to 10 ppm. These observations indicated that CoHA synthesized using both DC and PC power supplies had no obvious cytotoxicity, and PC_1_-CoHA significantly improved bone cell growth. Moreover, HA shows that the release of calcium and phosphorus ions contributes to cell growth. However, in the three sets of CoHA samples, a small number of cobalt ions added would give better cell growth than pure HA.

### 3.8. Extracellular Matrix Mineralization

Calcium deposition is one of the markers for osteoblast growth and differentiation, as well as for osteogenic potential. The MG63 cells were dyed using ARS after culturing in the CoHA extract solution for 7 days. Figure 10 shows the distribution of calcium ion deposition, observed using an optical microscope, which was used for evaluating cell growth and differentiation. The bright red area represents calcium ion deposition. PC_1_-CoHA and PC_2_-CoHA showed more calcium ion depositions on the extracellular matrix. The quantitative analysis results are shown in Figure 10H. After 7 days of culture, cell differentiation for DC-CoHA, PC_1_-CoHA, and PC_2_-CoHA was significantly higher than the other groups, and was 27–31% higher than the TCP control group. The culture of cells using extract solutions with cobalt ion concentrations of 1 ppm and 10 ppm showed that lower cobalt ion concentration could promote MG63 cell growth. A previous study found that cobalt ions can mimic the hypoxic environment and activate HIF-1α to promote VEGF production [64]. The differentiation of MG63 cells was probably enhanced under the influence of cobalt ions released by CoHA [65,66,67]. In addition, Nenad Ignjatovic et al. reported that hydrothermal synthesis of CoHA was implanted in the mandible of a rat model of osteoporosis, indicating that the mineral deposition rate of cobalt ion-filled sites is higher than that of pure HA and accelerates the rate of bone formation [15,21]. This is similar to the results of our study, indicating that cobalt ions contribute to the growth and differentiation of bone cells. 

### 3.9. Antibacterial

The filler material of the bone tissue defect, in addition to helping the growth and differentiation of the bone cells, must also be considered in terms of antibacterial ability. Through the inhibition of bacteria around the bone defect, it reduces the risk of infection to improve the success rate of surgery. In this work, *E. coli* is used as model organism since it is used frequently in antibacterial experiments. The bacteria was cultured with 6 mg/mL of three CoHA samples for 18 h and then antibacterial evaluation by culturing in BHI agar plates for 24 h and counting the number of colony forming units (CFU) as shown in Figure 11. Results show that all three groups of CoHA have excellent antibacterial activity (Figure 11A–D). Compared with the control group, PC_1_-CoHA (96%) had the highest sterilization rate, followed by DC-CoHA (95%) and PC_2_-CoHA (94%) (Figure 11E). ICP-OES analysis showed that PC1-CoHA had the highest release of cobalt ions after soaking in PBS for seven days (Table 3), so it had the best antibacterial ability. This can be explained by the fact that the antibacterial effect mainly comes from the release of cobalt ions. Other literature also reported that cobalt oxide and cobalt ions have good antibacterial performance [68].

## 4. Conclusions

CoHA was successfully prepared using electrochemical deposition and the effects of direct and pulse current supply modes on CoHA were discussed. A more uniform deposition of CoHA was obtained on a titanium surface using pulse electrochemical deposition. Through the direct current power mode, more cobalt ions attach to the surface, while the pulse current mode has higher cobalt ions in the interior. Magnetic analysis results show that more cobalt ions adhere to the surface of CoHA and are of smaller particle size, which will enhance the magnetic properties of the material. CoHA should reduce the inflammatory response of bone tissue after filling due to its free radical scavenging capability. MRI results showed that CoHA had a clear T_2_ image at low concentrations, and had a higher relaxation rate than other magnetic nanoparticles reported in literature; thus, it can be used potentially for localization in T_2_ imaging. The results of biocompatibility indicate that CoHA promotes the growth and differentiation of bone cells by releasing a small amount of cobalt ions. Furthermore, we emphasize on the application of PC_1_-CoHA as an available and potential bone filling material against *E. coli*. In conclusion, CoHA produced using pulse current has more deposition, higher degradation rate, better bioactivity and antibacterial ability than that produced using direct current. It can improve bone regeneration and induce bone growth, and can be applied in track and localization by MRI. It is expected to be helpful for the development of bone-filled materials.

## Figures and Tables

**Figure 1 materials-12-00116-f001:**
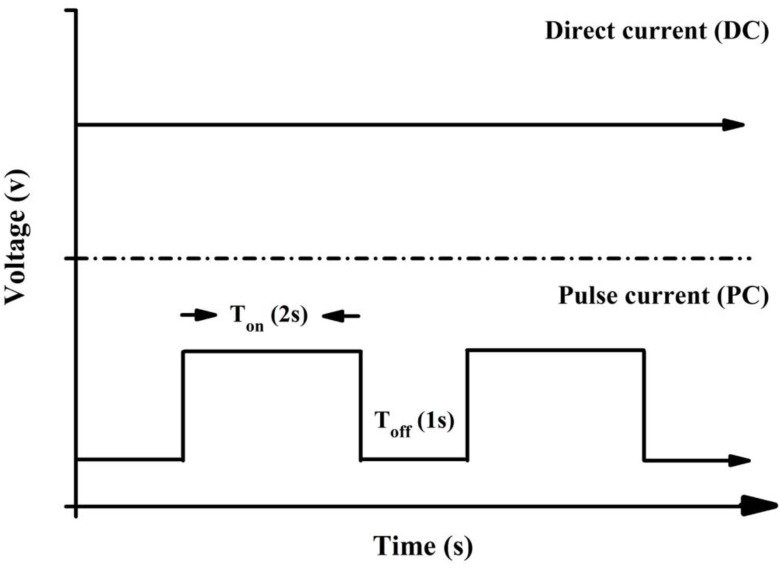
Schematic diagram of direct current and pulse current supply.

**Figure 2 materials-12-00116-f002:**
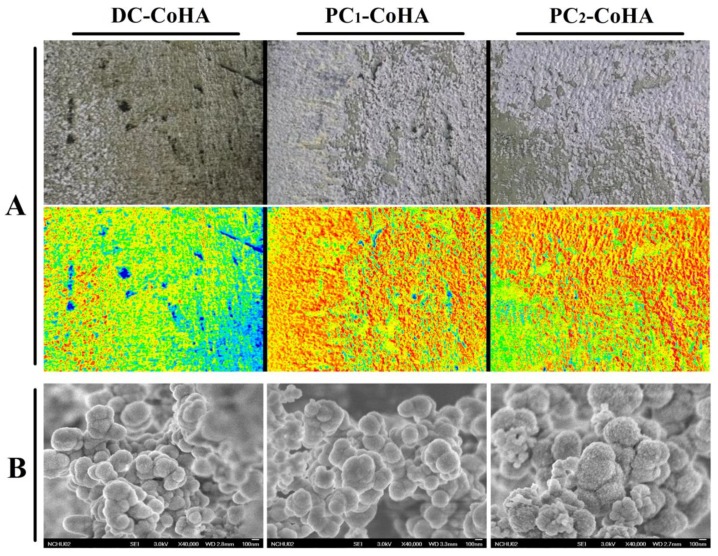
The surface morphology of the Co-HA particles was observed by (**A**) camera and (**B**) field-emission scanning electron microscope (FE-SEM). The color map is obtained using the surface image via Image J software analysis and red represents the distribution of Co-HA.

**Figure 3 materials-12-00116-f003:**
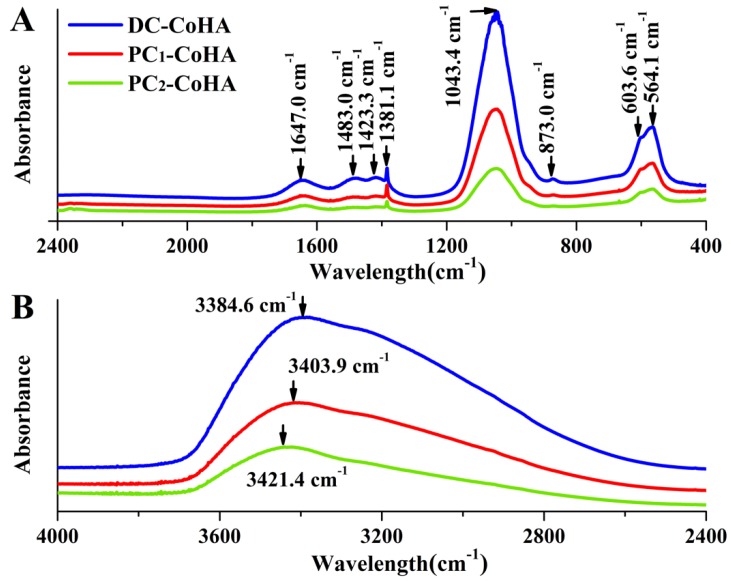
The attenuated total reflectance-Fourier transform infrared spectroscopy (ATR-FTIR) spectrum of the CoHA particles. (**A**) at 400–2400 cm^−1^ range, (**B**) at 2400–4000 cm^−1^ range.

**Figure 4 materials-12-00116-f004:**
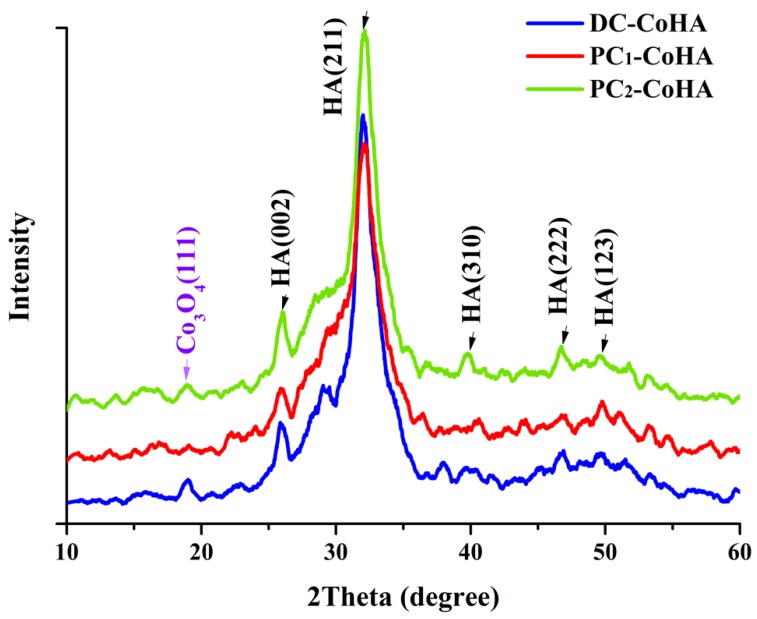
XRD patterns of the CoHA samples synthesized with different power supply.

**Figure 5 materials-12-00116-f005:**
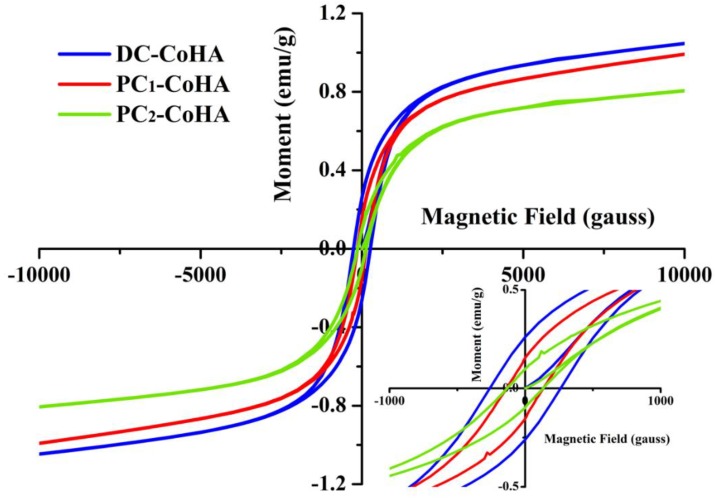
Mass magnetization measurements of Co-HA sample by superconducting quantum interference device (SQUID).

**Figure 6 materials-12-00116-f006:**
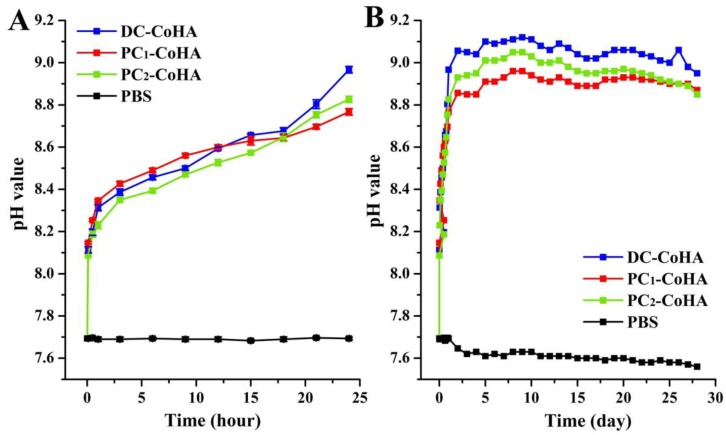
The pH values of different CoHA particles soaked in PBS solution at (**A**) 24 h and (**B**) 4 weeks.

**Figure 7 materials-12-00116-f007:**
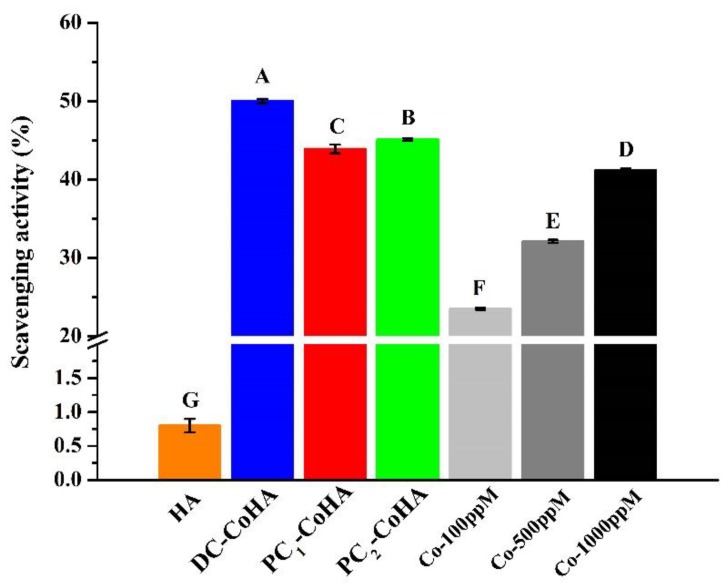
Free scavenging ratios of CoHA particles (*p* < 0.05).

**Figure 8 materials-12-00116-f008:**
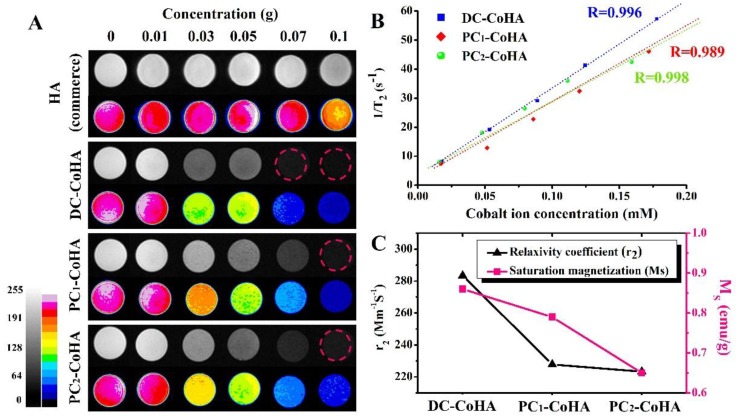
(**A**) T_2_-weighted magnetic resonance imaging (MRI) images of hydroxyapatite (HA) (commerce), DC-CoHA, PC_1_-CoHA and PC_2_-CoHA suspended in gelatinat at different concentrations; (**B**) the T_2_ relaxation rate *R*_2_ (1/*T*_2_) against cobalt concentration of CoHA; (**C**) relaxivity coefficient (*r*_2_) and saturation magnetization (Ms) of the relative relationship.

**Figure 9 materials-12-00116-f009:**
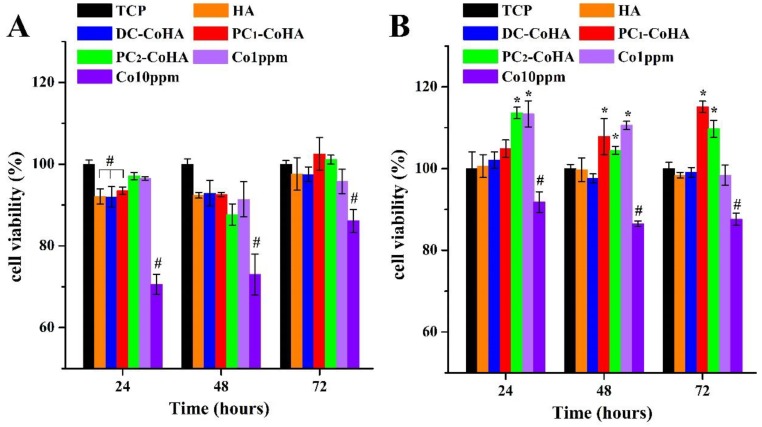
CoHA cytotoxicity test using (**A**) L929 cells and (**B**) MG63 cells, respectively. *: Significantly higher than the control group (TCP). #: Significantly lower than the control group (*p* < 0.01).

**Figure 10 materials-12-00116-f010:**
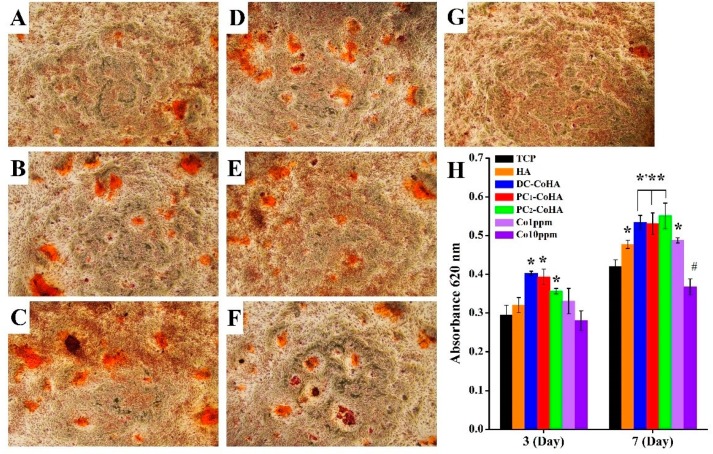
The degree of MG63 cells were cultured for 7 days with extracellular matrix mineralization determined by Alizarin Red S (Sigma) staining. (**A**) tissue culture plate (TCP), (**B**) HA, (**C**) DC-CoHA, (**D**) PC_1_-CoHA (**E**) PC_2_-CoHA, (**F**) Co1ppm and (**G**) Co10ppm. (**H**) Quantitative analysis of matrix deposition mineralization was performed at 3 and 7 days. *: Significantly higher than the control group (TCP) **: Significantly higher than HA and Co1ppm groups. #: Significantly lower than the control group (*p* < 0.001).

**Figure 11 materials-12-00116-f011:**
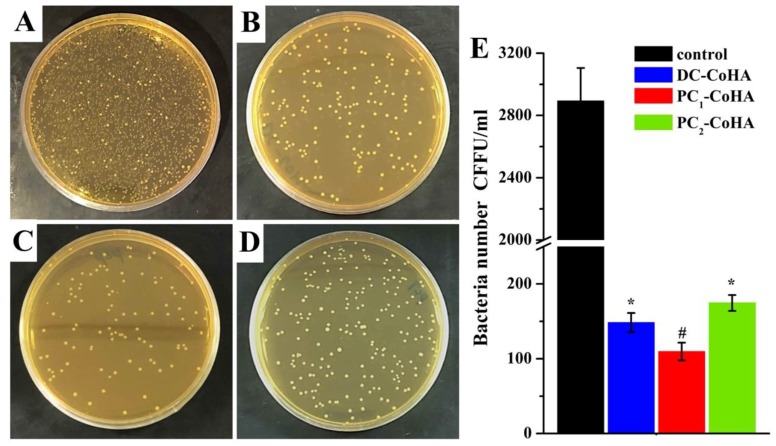
Images of the antibacterial results for relatively bacterial colonies on agar plates: (**A**) control (BHI), (**B**) DC-CoHA, (**C**) PC_1_-CoHA and (**D**) PC_2_-CoHA sample. (**E**) Antibacterial activity of the analyzed sample on *E. coli*. (*: significantly smaller than control group. #: significantly smaller than all the other samples, *p* < 0.05, mean ± standard deviation (SD), *n* = 4).

**Table 1 materials-12-00116-t001:** The particles size and magnetism parameters (*Hc*, *Ms* and *Mr*) of CoHA.

Sample	Particle Size			Magnetism Parameters		
	Length (nm)	Width (nm)	Aspect Ratio	*Hc* (Oe)	*Ms* (emu/g)	*Hr* (emu/g)
DC-CoHA	365 ± 11	285 ± 80	1.28	261.41	0.86	0.26
PC_1_-CoHA	418 ± 80	370 ± 90	1.13	133.73	0.79	0.15
PC_2_-CoHA	602 ± 73	510 ± 67	1.18	132.18	0.65	0.09

**Table 2 materials-12-00116-t002:** The construct parameters of the as synthesized CoHA particles by X-ray diffraction (XRD) analysis.

Sample	2θ	Line Width (FWHM)	Crystallite Size X_s_ (nm)	Fraction Crystallinity X_c_
DC-CoHA	25.87	0.77	1.849	0.030
PC_1_-CoHA	25.94	1.08	1.318	0.011
PC_2_-CoHA	25.92	0.91	1.565	0.018

**Table 3 materials-12-00116-t003:** Element ratios inside CoHA particles and ion release concentration in CoHA immersed phosphate buffer solution by inductively coupled plasma optical emission spectrometry (ICP-OES).

Sample	Particle		Extraction Solution		
	Ca + Co/P	Xco (%)	Ca (ppM)	Co (ppM)	P (ppM)
PBS	N.D.	N.D.	N.D.	N.D.	541
DC-CoHA	1.72	14.0	1.04	1.45	434
PC_1_-CoHA	1.67	14.0	1.68	1.86	429
PC_2_-CoHA	1.68	13.0	1.21	1.72	425

The extraction solution was obtained by immersed the powder in phosphate-buffered saline (PBS) at 37 °C for 7 days.

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
