# Peer review of "A Comparative Study on the Direct and Pulsed Current Electrodeposition of Cobalt-Substituted Hydroxyapatite for Magnetic Resonance Imaging Application"

_materials, 2018, doi:10.3390/ma12010116_

Round 1
Reviewer 1 Report
Authors have prepared Co-containing hydroxyapatite nanoparticles using different methods of electrodeposition. Materials were then characterized by a number of analytical methods, and their behaviour as potential MRI contrast agent was studied. Influence of nanoparticles on cell growth was studied.
The used language should be improved – in the present form it is sometimes little complicated to understand, what the authors have exactly meant by given sentences. However, I have also some objections towards factual merit of some data acquisition and interpretation of the results. Therefore, to my opinion, the manuscript cannot be published before major revision (and re-evaluation) will be made.
Some specific points:
Is there any explanation, why electrodeposition occurs only on cathode? Material contains also anions, so, in reverse case, some deposition can possibly occur also on anode. What has happened on anode? Deposition, dissolution of the anode material, nothing? Could iron be released from anode material?
Electrolyte contained 42 mM Ca and 7 mM Co (i.e. 6:1 ratio) according to experimental, but, in the Results (chapter 3.2), 10% content of Co in the electrolyte is mentioned. Why such discrepancy? In the view of 42/7, content of 13-14% in the final material is just about statistical, with no significant enrichment by Co.
It is meaningless to discuss Ca/P ratio as a reason for faster dissolution of Co/dopped HA, as it is not metal deficient structure. Ca+Co/P ratio should be taken into account, which is close to theoretical value, i.e. no metal deficiency was observed.
What method was used in treatment of Fig. 2 for colorization of Co-content? Please specify, how exactly Co-content on the surface/interior the material was determined. Is Co on surface really elemental cobalt?
How fraction of crystallinity and crystallite size were determined?
Why carbonate is present in the material? There was no source of CO2 to obtain such material, and C=O vibrations are relatively intense.
It is rather unusual, that mostly amorphous material (according to presented level of crystallinity) show magnetic hysteresis. Why? Relevant discussion should be somewhat extended – more information about materials from literature should be given (maybe Table? – exact composition, size, crystallinity, Ms, Hr, Hc...).
A claim that PC1-CoHA is decomposed/dissolved faster than other samples based on one measurement of Co concentration is too presumptuous. Several data points in different times should be acquired to have support for such statement about kinetics. Especially when pH-equilibrium was reached after much shorter time (1-2 days) comparing to time when Co content was analyzed (and maybe, the suspension was already equilibrated).
It is not clear if effect on cell growth is caused only by released Co, or also by increase of concentration of Ca and phosphate. HA-containing medium should be used for control.
Is scavenging of radicals caused by solid material, or by presence of free paramagnetic Co ions?
Why relaxivities r2 of the materials so differ, but slopes of lines in Fig. 8 are more-less the same? Why the data with poorest linearity have highest agreement factor R2?
Is any explanation, why relaxivity r2 is so high, although the material is mainly amorphous and therefore cannot show superparamagnetic/ferromagnetic interaction? It seems to be inconsistent with presented low crystallinity of the material.
Author Response
Response to Reviewer 1 Comments
Thanks for your suggestion. We will to English editing of manuscript and answer your question.
Q1. Is there any explanation, why electrodeposition occurs only on cathode? Material contains also anions, so, in reverse case, some deposition can possibly occur also on anode. What has happened on anode? Deposition, dissolution of the anode material, nothing? Could iron be released from anode material?
A1: In the electrochemical reaction, the anode is release electrons and cause electrode plate corrosion such as titanium dioxide nanotubes formation. At the same time, the cathode receives electrons and promotes ion reduction. Then deposited on the plate [1, 2]. In addition, the synthesized cobalt-substituted hydroxyapatite (CoHA) was confirmed by elemental analysis (EDS and ICP/OES) and confirm the absence of other metal ions.
Anode = M→Mn++ne−
Cathode = Mn++ne−→M.
Figure S1. elemental analysis of three kind of CoHA by EDS. (Please see the attachment)
Q2. Electrolyte contained 42 mM Ca and 7 mM Co (i.e. 6:1 ratio) according to experimental, but, in the Results (chapter 3.2), 10% content of Co in the electrolyte is mentioned. Why such discrepancy? In the view of 42/7, content of 13-14% in the final material is just about statistical, with no significant enrichment by Co.
A2: Because the initial setting of co should be 10% of the whole. The calculation formula is as follows:
10.5%
ICP-OES is used to quantify the overall elemental composition of the powder, including calcium, phosphorus and cobalt ions (Table S1). After calculation, the total content of cobalt ions is DC-CoHA (14%), PC1-CoHA (14%) and PC2-CoHA (13%), respectively.
Table S1. The overall elemental composition of the powder. | |||
Sample | DC-CoHA | PC1-CoHA | PC2-CoHA |
Ca (mg/L) | 251.5 | 238 | 231.8 |
P (mg/L) | 145.2 | 142 | 134.7 |
Co (mg/L) | 104.8 | 101.3 | 89.5 |
Q3. It is meaningless to discuss Ca/P ratio as a reason for faster dissolution of Co/dopped HA, as it is not metal deficient structure. Ca+Co/P ratio should be taken into account, which is close to theoretical value, i.e. no metal deficiency was observed.
A3: Thanks for your suggestion. This sentence will be deleted in this article (Page 6, Line 198).
Q4. What method was used in treatment of Fig. 2 for colorization of Co-content? Please specify, how exactly Co-content on the surface/interior the material was determined. Is Co on surface really elemental cobalt?
A4: In our original text: “The distribution of apatite deposits was analyzed using an imaging software (ImageJ, USA), indicated by the red area observed in Fig. 2A” (Page 5, Line 187). It's mean that CoHA is deposited on titanium. The color map of surface image is obtained using ImageJ image software and red represents the distribution of CoHA. On the other hand, our study detects the elemental composition inside and on the surface of the CoHA by EDS and ICP/OES, respectivity. It was confirmed that the surface of the powder and the interior contained cobalt.
Q5. How fraction of crystallinity and crystallite size were determined?
A5: We add a description about calculate of crystallinity and crystal size in the manuscript (Page 3, Line 103). Describe the sentence below “The hydroxyapatite crystal size is calculated by the following formula, Xs= 0.9λ/ FWHM cosθ, Xsis the grain size (nm), λ is the CuKα radiation, FWHM (Full Width at Half Maximum) is the half width of the diffraction peak, θ for the diffraction peak angle. The crystallinity calculation is calculated by the following formula Xc= (K/β)3, K is a constant of 0.24[3].”
Q6. Why carbonate is present in the material? There was no source of CO2to obtain such material, and C=O vibrations are relatively intense.
A6: In our study, electrochemical deposition is carried out in the atmosphere. Therefore, air (contain of carbon dioxide) is absorbed in electrolyte solution during the reaction. Previous literatures have reported that and C=O vibrations have also appeared in CoHA using via ion exchange and wet synthesis (Figure S2.). The authors also explain that this phenomenon is the production of additional peaks in the synthesis process[3]. This situation also appears in the literature on the preparation of HA using electrochemical deposition (Figure S3.)[4].
Figure S2. FT-IR spectra for HA, CoHA synthesized via ion exchange, and CoHA synthesized via wet synthesis. Labels on the HA spectrum also apply to the identical peaks in the CoHA spectra[3].(Please see the attachment)
Figure S3. FT-IR spectra of the sample deposited with 2.5 mA/cm2current density[4]. (Please see the attachment)
Q7. It is rather unusual, that mostly amorphous material (according to presented level of crystallinity) show magnetic hysteresis. Why? Relevant discussion should be somewhat extended – more information about materials from literature should be given (maybe Table? – exact composition, size, crystallinity, Ms, Hr, Hc...).
A7: There are still a few amorphous materials that still have magnetic hysteresis. For example, Erica Kramer et al. used ion exchange and wet synthesis to synthesize CoHA. The authors point out that the two CoHAs have very low crystallinity due to the addition of Co and it shows that they belong to the amorphous material[3]. However, the results of the magnetic analysis showed that the magnetization of the ion exchanged and wet-synthesized CoHA showed a positive slope indicating a paramagnetic property (Figure S4). Therefore, although the crystallinity of the material is very low in this study, CoHA is superparamagnetic due to the presence of cobalt ions. Many thanks to reviewer for the question and we also add this discussion to this article (Page 9, Line 261).
Figure S4. Mass magnetization measurements of HA, CoHA via ion exchange, and CoHA via wet synthesis [3]. (Please see the attachment)
Q8. A claim that PC1-CoHA is decomposed/dissolved faster than other samples based on one measurement of Co concentration is too presumptuous. Several data points in different times should be acquired to have support for such statement about kinetics. Especially when pH-equilibrium was reached after much shorter time (1-2 days) comparing to time when Co content was analyzed (and maybe, the suspension was already equilibrated).
A8: We show the results of the short-term dissolution test in Fig. 6(A). The results showed that when three kinds of CoHA were added to the PBS, respectively. The release of phosphorus ions caused a significant increase in pH. In addition, the pH rises a lot before 24 hours, followed by a slow rise. Interestingly, the pH of the original PC1-CoHA was up-regulated within 24 hours, but it changed at 18 hours. We explain that when the powder is exposed to PBS, the phosphorus ion (pH value up) is first released and then calcium and cobalt ions (pH value down). Consequently, when the pH value changes, it means that there are more calcium and cobalt ions released. Therefore, CoHA prepared by the PC method can obtain a good degradation effect compared with the DC method. In particular, PC1-CoHA has a significant difference at 24 hours. The above discussion was added to the manuscript (Page 9, Line 274).
Fig. 6. The pH values of different CoHA particles soaked in PBS solutionat (A) 24 hours and (B) 4 weeks. (Please see the attachment)
Q9. It is not clear if effect on cell growth is caused only by released Co, or also by increase of concentration of Ca and phosphate. HA-containing medium should be used for control.
A9: We are very grateful to the advice given by reviewer. The design of the original experiment has hydroxyapatite (HA)-containing medium as a control group. In this article, we have appended the control group of HA. The results of HA show that the release of calcium and phosphorus ions contributes to cell growth. However, in the three kinds of CoHA, a small number of cobalt ions added would give better cell growth than HA. In the literature, when hydrothermal sunthesis CoHA was implanted in the mandible of a rat model of osteoporosis, The mineral deposition rate of cobalt ion-filled sites is higher than that of Hap and accelerates the rate of bone formation [5, 6]. Above result is similar to the results of our experiments, indicating that cobalt ions contribute to the growth and differentiation of bone cells. The above discussion was added to the manuscript (Page12, Line 360 and Page13, Line 379).
Fig. 9. CoHA cytotoxicity test using (A) L929 cells (B) MG63 cells, respectively. *: Significantly higher than the control group (TCP) #: Significantly lower than the control group (p<0.01).< p=""> (Please see the attachment)
Fig. 10. The degree of MG63 cells were cultured for 7 days extracellular matrix mineralization was determined by Alizarin Red S (Sigma) staining. (A)TCP, (B)HA, (C) DC-CoHA, (D) PC1-CoHA (E) PC2-CoHA, (F) Co1ppm and (G) Co10ppm. (H) Quantitative analysis of matrix deposition mineralization was performed at 3 and 7 days. *: Significantly higher than the control group (TCP) **: Significantly higher than HA and Co1ppm groups #: Significantly lower than the control group (p<0.001).< p=""> (Please see the attachment)
Q10. Is scavenging of radicals caused by solid material, or by presence of free paramagnetic Co ions?
A10: We added commercial HA and Co standard solution as a control group. The results show that the resulting free radical scavenging does come primarily from Co ions.
Fig. 7. Free Scavenging ratios of CoHA particles (p < 0.05).
Q11. Why relaxivities r2of the materials so differ, but slopes of lines in Fig. 8 are more-less the same? Why the data with poorest linearity have highest agreement factor R2?
A11: The cause of Figure 8 is mainly the concentration ratio, which is equivalent to the number of target molecules contained in the sample. The influence of quantity is much larger than the difference between different types, so it seems that the concentration affects the slope.
Q12. Is any explanation, why relaxivity r2is so high, although the material is mainly amorphous and therefore cannot show superparamagnetic/ferromagnetic interaction? It seems to be inconsistent with presented low crystallinity of the material.
A12: It is mentioned in the article that the surface of DC-CoHA may be attached with more cobalt ions (with unpaired electrons, easy to disturb the magnetic field), produce in higher saturation magnetization (Ms), resulting in higher relaxivity r2than PC1-CoHA and PC2-CoHA(Fig. 8C). At the same time, compared with the literature, it is found that relaxivity r2is proportional to Ms [5, 6]. Therefore DC-CoHA has a higher relaxivity r2. Another possibility is the size of the crystalline molecule. Generally speaking, the larger the crystal size, the higher the r2value. Many thanks to Review for the question, and we also add this discussion to this article (Page 11, Line 332). We also created a new table to illustrate the relationship between relaxation rates and magnetic properties.
Table S2. Relaxivity coefficient (r2) and saturation magnetization (Ms) of the relative relationship. | |||
Sample code | Ms (emug-1) | r2 (mM-1s-1) | references |
Co-Fe | 55.16 | 110.9 | [5] |
60.59 | 169.9 | ||
64.29 | 301.8 | ||
Fe3O4 | 215 | 644 | [6] |
101 | 218 | ||
80 | 130 | ||
43 | 106 | ||
25 | 78 | ||
DC-CoHA | 0.86 | 283.4 | |
PC1-CoHA | 0.79 | 227.8 | |
PC2-CoHA | 0.65 | 223.3 |
References
1. Parcharoen, Y.; Kajitvichyanukul, P.; Sirivisoot, S.; Termsuksawad, P., Hydroxyapatite electrodeposition on anodized titanium nanotubes for orthopedic applications. Applied Surface Science 2014, 311, 54-61.
2. KUO-HSIUNG, C., Dental materials science 2004.
3. Kramer, E.; Itzkowitz, E.; Wei, M., Synthesis and characterization of cobalt-substituted hydroxyapatite powders. Ceramics International 2014, 40, (8), 13471-13480.
4. He, D.-H.; Wang, P.; Liu, P.; Liu, X.-K.; Ma, F.-C.; Zhao, J., HA coating fabricated by electrochemical deposition on modified Ti6Al4V alloy. Surface and Coatings Technology 2016, 301, 6-12.
5. Joshi, H. M.; Lin, Y. P.; Aslam, M.; Prasad, P.; Schultz-Sikma, E. A.; Edelman, R.; Meade, T.; Dravid, V. P., Effects of shape and size of cobalt ferrite nanostructures on their MRI contrast and thermal activation. The Journal of Physical Chemistry C 2009, 113, (41), 17761-17767.
6. Jun, Y.-w.; Huh, Y.-M.; Choi, J.-s.; Lee, J.-H.; Song, H.-T.; Kim, S.; Kim, S.; Yoon, S.; Kim, K.-S.; Shin, J.-S., Nanoscale size effect of magnetic nanocrystals and their utilization for cancer diagnosis via magnetic resonance imaging. Journal of the American Chemical Society 2005, 127, (16), 5732-5733.

Reviewer 2 Report
In this study, authors explain the use of Cobalt ions as hypoxia-inducible factors and accelerate bone repair. At the same time, paramagnetic of Cobalt ions allows the study of medical imaging and target drugs. Authors synthesize cobalt-substituted hydroxyapatite (CoHA) via electrochemical deposition. The effects of different current supply modes on the morphology, magnetic properties and biodegradability were also compared in this study.
This manuscript is well written and the characteristics of CoHA are carefully examined in this study.
I recommend publication of this manuscript without any changes.
Author Response
Response to Reviewer 2 Comments
This manuscript is well written and the characteristics of CoHA are carefully examined in this study. I recommend publication of this manuscript without any changes.
Answer: Thanks for your reply.
Reviewer 3 Report
The paper is suitable for the acceptance after English editing.
Author Response
Response to Reviewer 3 Comments
The paper is suitable for the acceptance after English editing.
Answer: Thanks for your suggestion. We will to English editing of manuscript.
Round 2
Reviewer 1 Report
Language should be improved. Beside this, I have still some questions, related to the previous points and to author's answers:
A1: In the electrochemical reaction, the anode is release electrons and cause electrode plate corrosion such as titanium dioxide nanotubes formation. At the same time, the cathode receives electrons and promotes ion reduction. Then deposited on the plate [1, 2]. In addition, the synthesized cobalt-substituted hydroxyapatite (CoHA) was confirmed by elemental analysis (EDS and ICP/OES) and confirm the absence of other metal ions.
Anode = M→Mn++ne−
Cathode = Mn++ne−→M.
Figure S1. elemental analysis of three kind of CoHA by EDS. (Please see the attachment)
Q1-1: In electrolysis, polarity of electrodes is reversed compared to the galvanic couple. Anode is positive and receives electrons, and cathode negative and releases electrons. There is still confusion – in the manuscript, authors say that Ti was a cathode, but in the answer A1 they mentioned that Ti is an anode...
A2: Because the initial setting of co should be 10% of the whole. The calculation formula is as follows:
10.5%
ICP-OES is used to quantify the overall elemental composition of the powder, including calcium, phosphorus and cobalt ions (Table S1). After calculation, the total content of cobalt ions is DC-CoHA (14%), PC1-CoHA (14%) and PC2-CoHA (13%), respectively.
Table S1. The overall elemental composition of the powder. | |||
Sample | DC-CoHA | PC1-CoHA | PC2-CoHA |
Ca (mg/L) | 251.5 | 238 | 231.8 |
P (mg/L) | 145.2 | 142 | 134.7 |
Co (mg/L) | 104.8 | 101.3 | 89.5 |
Q2-1: How was calculated percentage of Co based on the results presented in Table S1 (see above)? What means 14 % of Co? Molar %? Weight %? What is 100 %?
In Experimental, authors say that:
The electrolytic solution used 42mM calcium nitrate (Shimada chemical works, Tokyo, Japan) and 25mM ammonium dihydrogen phosphate (Showa chemical industry, Tokyo, Japan) in deionized water [34]. The pH of electrolyte was adjusted at 3.52 using hydrochloric acid and tris(hydroxymethyl)aminomethane (Tris). Finally, added 7.9 mM Cobalt chloride (Shimada chemical works, Tokyo, Japan) to stir until completely dissolved.
Q2-2: How a value of 10.5 % is calculated from this dataset? Obviously, total metal to phosphate ratio is 2:0, with Ca:Co about 5:1...
A8: We show the results of the short-term dissolution test in Fig. 6(A). The results showed that when three kinds of CoHA were added to the PBS, respectively. The release of phosphorus ions caused a significant increase in pH. In addition, the pH rises a lot before 24 hours, followed by a slow rise. Interestingly, the pH of the original PC1-CoHA was up-regulated within 24 hours, but it changed at 18 hours. We explain that when the powder is exposed to PBS, the phosphorus ion (pH value up) is first released and then calcium and cobalt ions (pH value down). Consequently, when the pH value changes, it means that there are more calcium and cobalt ions released. Therefore, CoHA prepared by the PC method can obtain a good degradation effect compared with the DC method. In particular, PC1-CoHA has a significant difference at 24 hours. The above discussion was added to the manuscript (Page 9, Line 274).
Fig. 6. The pH values of different CoHA particles soaked in PBS solutionat (A) 24 hours and (B) 4 weeks. (Please see the attachment)
Q8-1: The text is very speculative and confusing – why only anion should be released (and no cation) in the beginning, and cations are released after 24 h...? It is against principle of electroneutrality. Moreover, if only phosphate was released, some hydroxide must be absorbed, and, thus, leaving H+ in the solution and dropping down pH...
A11: The cause of Figure 8 is mainly the concentration ratio, which is equivalent to the number of target molecules contained in the sample. The influence of quantity is much larger than the difference between different types, so it seems that the concentration affects the slope.
Q11-1 I still do not understand, how presented values 283.4mM-1s-1 etc. were calculated, and why differ so much for materials, if data-points lie on the same slope. Especially, red andgreen lines in Fig. 8 are more-less the same, except of one green data-point. So, was relaxivity value calculated from one point with the highest concentration< What is meaning of agreement factors R2, id no linear regression is shown? Why is the R2 closest to 1 for the green dataset, which has obviously the worst linearity?
A12: It is mentioned in the article that the surface of DC-CoHA may be attached with more cobalt ions (with unpaired electrons, easy to disturb the magnetic field), produce in higher saturation magnetization (Ms), resulting in higher relaxivity r2than PC1-CoHA and PC2-CoHA(Fig. 8C). At the same time, compared with the literature, it is found that relaxivity r2is proportional to Ms [5, 6]. Therefore DC-CoHA has a higher relaxivity r2. Another possibility is the size of the crystalline molecule. Generally speaking, the larger the crystal size, the higher the r2value. Many thanks to Review for the question, and we also add this discussion to this article (Page 11, Line 332). We also created a new table to illustrate the relationship between relaxation rates and magnetic properties.
Table S2. Relaxivity coefficient (r2) and saturation magnetization (Ms) of the relative relationship. | |||
Sample code | Ms (emug-1) | r2 (mM-1s-1) | references |
Co-Fe | 55.16 | 110.9 | [5] |
60.59 | 169.9 | ||
64.29 | 301.8 | ||
Fe3O4 | 215 | 644 | [6] |
101 | 218 | ||
80 | 130 | ||
43 | 106 | ||
25 | 78 | ||
DC-CoHA | 0.86 | 283.4 | |
PC1-CoHA | 0.79 | 227.8 | |
PC2-CoHA | 0.65 | 223.3 |
Q12-1: I have still some doubts about attribution of the r2-effect to the surface content of Co2+ ions. Is the r2-relaxation caused by dissolved Co2+ ions? In (pseudo)crystalline materials, transversal relaxivity is caused predominantly by large magnetic effect of remanent magnetic momentum of ferromagnetic lattice, and no dissolution of paramagnetic ions is needed. What is r2 of free Co2+ ion?
Author Response
Response to Reviewer 1 Comments
Q1-1: In electrolysis, polarity of electrodes is reversed compared to the galvanic couple. Anode is positive and receives electrons, and cathode negative and releases electrons. There is still confusion – in the manuscript, authors say that Ti was a cathode, but in the answer A1 they mentioned that Ti is an anode.
A1-1: Thanks to the reviewer’s comment. Original sentence in manuscript is “Titanium piece as a cathode, 304 stainless steel plate as an anode” (Page 2, Line 83). It is very clear that the titanium plate acts as a cathode in the electrochemical deposition process. On the other hand, previously reply A1 is described that titanium sheet as anode in anodic oxidation. Then titanium dioxide nanotubes formed in anode surface.
Q2-1: How was calculated percentage of Co based on the results presented in Table S1 (see above)? What means 14 % of Co? Molar %? Weight %? What is 100 %?
Table S1. The overall elemental composition of the powder. | |||
Sample | DC-CoHA | PC1-CoHA | PC2-CoHA |
Ca (mg/L) | 251.5 | 238 | 231.8 |
P (mg/L) | 145.2 | 142 | 134.7 |
Co (mg/L) | 104.8 | 101.3 | 89.5 |
A2-1:The overall elemental composition of cobalt-substituted hydroxyapatite is obtained by ICP/OES. The unit of concentration is mg per L (see table S1). Then weight of element can convert to mole number of elements and bring the formula (eq1) to calculate the mole fraction (molar %) of cobalt in all elements. ).(eq 1). The 100% is sum of mole fraction of elements.
Q2-2: How a value of 10.5 % is calculated from this dataset? Obviously, total metal to phosphate ratio is 2:0, with Ca:Co about 5:1.
A2-2:In our study, the composition of electrolyte is expressed in terms of molarity (see Table S2). The value of 10.5% is obtained by dividing the concentration of cobalt by the concentration of overall (caclium, cobalt and phosphate). In the other hand, sum of metal (Ca, Co) concentration divided by phosphate concentration is 1.996 similar 2.0 and calcium: cobalt = 5.3: 1.
Table S2. electrolyte composition | |||
Ca (mol/L) | PO4 (mol/L) | Co (mol/L) | Co % ) |
0.042 | 0.025 | 0.0079 | 10.5% |
Q8-1: The text is very speculative and confusing – why only anion should be released (and no cation) in the beginning, and cations are released after 24 h? It is against principle of electroneutrality. Moreover, if only phosphate was released, some hydroxide must be absorbed, and, thus, leaving H+in the solution and dropping down pH.
A8-1: The anion and cation will be released in the initial period. However, anion (phosphorus) is faster release than cation (calcium, cobalt) lead to increase of pH value. Then cation release slow down increase of pH value. This a tendency is consistent with the literature [1]. Slow dissolution of hydroxyapatite and increase of hydroxy group on solution lead to increase of pH value. In the other hand, considering that the ion concentration in the phosphate buffer solution may affect the dissolution of cobalt-substituted hydroxyapatite. The PBS are exchange to deionized water. The result is similar to above finding (see Figure S2). The above described is added on article (Page 10, Line 277).
Figure S1.pH curve of the cobalt ion exchange procedure at room temperature [1]. (Please see the attachment)
Figure S2. The pH value of ions in deionized water at room temperature. (Please see the attachment)
Q11-1 I still do not understand, how presented values 283.4mM-1s-1etc. were calculated, and why differ so much for materials, if data-points lie on the same slope. Especially, red andgreen lines in Fig. 8 are more-less the same, except of one green data-point. So, was relaxivity value calculated from one point with the highest concentration< What is meaning of agreement factors R2, id no linear regression is shown? Why is the R2closest to 1 for the green dataset, which has obviously the worst linearity?
A11-1: The calculation of the relaxation coefficient (r2) is mentioned in this article (Page 4, Line 139). The r2 is calculated by the following formula (eq 2)[2]:
(eq 2). where is the observed relaxation time in the presence of CoHA, is the relaxation rate of pure gelatin and is theconcentration of cobalt ion. It can be known from the formula that the factors affecting the r2value are the T2value and the concentration of cobalt ions. When ceramic weight as the x-axis, the trendline is linearity (Figure S3 (A)). However, it is not the correct expression. Because cobalt content of each sample is different. Therefore, the concentration of cobalt as the X-axis, via linear regression, trendline looks more linearity (Figure S3 (B)).
Figure S3. T2relaxation rate R2(1/T2) against (A) ceramic weight and (B) cobalt concentration of Co-HA.
Table S2. Relaxivity coefficient (r2) and saturation magnetization (Ms) of the relative relationship. | |||
Sample code | Ms (emug-1) | r2 (mM-1s-1) | references |
Co-Fe | 55.16 | 110.9 | [2] |
60.59 | 169.9 | ||
64.29 | 301.8 | ||
Fe3O4 | 215 | 644 | [3] |
101 | 218 | ||
80 | 130 | ||
43 | 106 | ||
25 | 78 | ||
DC-CoHA | 0.86 | 283.4 | |
PC1-CoHA | 0.79 | 227.8 | |
PC2-CoHA | 0.65 | 223.3 |
Q12-1: I have still some doubts about attribution of the r2-effect to the surface content of Co2+ions. Is the r2-relaxation caused by dissolved Co2+ions?
In (pseudo)crystalline materials, transversal relaxivity is caused predominantly by large magnetic effect of remanent magnetic momentum of ferromagnetic lattice, and no dissolution of paramagnetic ions is needed. What is r2of free Co2+ion?
A12-1: Because of the presence of cobalt, CoHA have superparamagnetic performance. In addition, surface of DC-CoHA attached with more cobalt than PC1-CoHA and PC2-CoHA (with unpaired electrons and easy to disturbance the magnetic field), resulting in a higher saturation magnetization (Table S2) [4]. Not the effect of cobalt ion dissolution. In the other hand, the size of the crystal cluster also affects the relaxation rate r2 [4, 5]. Large crystal cluster have higher saturation magnetization and relaxation rate (Figure S5). Above result show the magnetization is proportional to the relaxation rate. Therefore, DC-CoHA has a high magnetization and high relaxation rate than another sample.
Figure S5. Surface spin canting effect of a nanoparticle upon magnetization (Mmagnetic moment, H external magnetic field). b–e) Nano-scale size effects of Fe3O4(MEIO) nanoparticles on magnetism and MR contrast effects. b) Transmission electron microscopic (TEM) images of 4, 6, 9, and 12 nm sized MEIO nanoparticles. c) Mass magnetization values, d) T2-weighted MR images (top: black and white, bottom: color), and e) relaxivity coefficient r2of the nano-particles presented in (a) [4].
References
1. Kramer, E.; Itzkowitz, E.; Wei, M., Synthesis and characterization of cobalt-substituted hydroxyapatite powders. Ceramics International 2014, 40, (8), 13471-13480.
2. Joshi, H. M.; Lin, Y. P.; Aslam, M.; Prasad, P.; Schultz-Sikma, E. A.; Edelman, R.; Meade, T.; Dravid, V. P., Effects of shape and size of cobalt ferrite nanostructures on their MRI contrast and thermal activation. The Journal of Physical Chemistry C 2009, 113, (41), 17761-17767.
3. Jun, Y.-w.; Huh, Y.-M.; Choi, J.-s.; Lee, J.-H.; Song, H.-T.; Kim, S.; Kim, S.; Yoon, S.; Kim, K.-S.; Shin, J.-S., Nanoscale size effect of magnetic nanocrystals and their utilization for cancer diagnosis via magnetic resonance imaging. Journal of the American Chemical Society 2005, 127, (16), 5732-5733.
4. Jun, Y. W.; Lee, J. H.; Cheon, J., Chemical design of nanoparticle probes for high-performance magnetic resonance imaging. Angew Chem Int Ed Engl 2008, 47, (28), 5122-35.
5. Shokrollahi, H., Contrast agents for MRI. Materials Science and Engineering: C 2013, 33, (8), 4485-4497.
